# Strong Fitness Costs of Fall Armyworm Resistance to Dual-Gene Bt Maize Are Magnified on Less-Suitable Host-Crop Cultivars

Oscar F. Santos-Amaya [1,*], Clébson S. Tavares [2,3], João Victor C. Rodrigues [3], Eugênio E. Oliveira [3], Raul Narciso C. Guedes [3] and Eliseu José G. Pereira [3,4,*]

1 Department of Agronomy, Universidad de Pamplona, Pamplona 543057, Colombia
2 Department of Entomology, University of Florida, Gainesville, FL 32611, USA; clebsondossantos@ufl.edu
3 Department of Entomology, Universidade Federal de Viçosa, Vicosa 36570-900, Brazil; jv.canazart@gmail.com (J.V.C.R.); eugenio@ufv.br (E.E.O.); guedes@ufv.br (R.N.C.G.)
4 National Institute of Science and Technology in Plant-Pest Interactions, Universidade Federal de Viçosa, Vicosa 36570-900, Brazil
* Correspondence: oscar.santos12@unipamplona.edu.co (O.F.S.-A.); eliseu.pereira@ufv.br (E.J.G.P.)

**Abstract:** We examined the fitness costs of resistance to transgenic Cry1A.105+Cry2Ab2 *Bacillus thuringiensis* (Bt) maize in the fall armyworm, *Spodoptera frugiperda*, a globally invasive pest species. Using Bt-resistant and -susceptible populations of similar genetic backgrounds, we compared insect life-history traits and population growth rates on the foliage of Bt or non-Bt cultivars of maize, soybean, and cotton. We found that (i) the resistance alleles led to a major reduction in insect fitness on seven of the eight cultivars of three host crops studied; (ii) developmental time was the life-history trait that accounted for most of the fitness variation of the armyworm, and (iii) the magnitude of fitness reduction of the resistant individuals was stronger on cotton foliage, on which the insects did not pass the resistance alleles to subsequent generations. These results show that fall armyworm resistance to Cry1A.105+Cry2Ab2-expressing Bt maize comes with strong fitness costs, which were magnified on less-suitable host plants for the insects. Thus, natural selection may help maintain or even restore the insect susceptibility to the pesticidal proteins and perhaps is a significant factor helping manage fall armyworm resistance to Cry1A.105+Cry2Ab2 Bt maize. These findings indicate that fitness costs of insect resistance to multi-toxin Bt crops can be strong, and host plants or conditions that magnify the fitness differential of susceptible insects could be specifically deployed with refuge to improve resistance management to Bt crops.

**Keywords:** pyramided Bt crops; Cry1/Cry2 toxins; fall armyworm; life-history traits; population fitness; phenotypic plasticity; resistance management

## 1. Introduction

Transgenic crops producing pesticidal proteins (or toxins) from the soil bacterium *Bacillus thuringiensis* (Bt) are an integral tool in the management of agricultural insect pests, adopted globally in more than 100 million ha [1,2]. As a state-of-the-art agrobiotechnology, Bt crop cultivars generally have remarkably high yield potential and high specificity to target pests, warranting their perceived value as economically and environmentally beneficial [1,3]. Nevertheless, some of these benefits are diminished as pests evolve resistance [4,5]. There are several cases of field-relevant resistance to specific Bt traits across major target pest species [5].

Some crop management strategies are recommended to delay the faster-than-expected development of Bt resistance in pest populations [6]. Bt crop cultivars are generally designed to produce a high dose of insecticidal protein in plant tissues; only very few resistant insects are expected to survive, producing susceptible offspring when mating with susceptible insects from a nearby non-Bt area (referred to as refuge) [6]. Additionally, modern Bt

crop cultivars may carry pyramided traits by producing pesticidal Bt proteins of different biochemical sites of action in the target insect midgut epithelia [7]. Nevertheless, the risk of losing pest susceptibility to Bt traits continues as insect pest populations can evolve resistance to Bt-pyramided crops [8–12]. The problem may occur faster than expected when resistance is not a recessive trait and has no disadvantage to the resistant insect in the absence of the toxicant, unfavorable conditions to maintain the pest susceptibility [13]. The competence to manage resistance evolution can be improved by better understanding its evolutionary ecology [4,6,14]. More specifically, data on the relative fitness of resistant and susceptible near-isogenic target insects on near-isogenic Bt and non-Bt crop cultivars and its inheritance [15–17] are valuable to inform resistance management programs [1,18,19].

The fall armyworm, *Spodoptera frugiperda* (Lepidoptera: Noctuidae), is a model species to study speciation, migration, and the evolutionary ecology of resistance to dual-gene Bt crops. This highly polyphagous herbivore exhibits host-plant-associated races and is the primary migratory maize pest in North America [20,21]. The fall armyworm was introduced in Africa in the last decade, later invading Asia and Oceania [22–28]. The armyworm is notorious for holding several cases of field-relevant resistance to single-gene Bt maize [29–33]. Given the species' remarkable capacity for adaption to population control measures [34], the gene pyramiding strategy in Bt crops to preserve fall armyworm susceptibility to Bt traits could be more efficacious if resistance to multi-Bt-trait cultivars comes at a high fitness cost to the individuals [9,17,35–37].

Fitness costs affect resistance evolution by selecting against resistant individuals in the absence of the selecting agent, where the individuals carrying resistance alleles have lower fitness than susceptible individuals [4,17,37]. Fitness costs of resistance to Bt crops can delay resistance by selecting against resistant insects in natural or structured refuges planted near the Bt crop and can even reverse resistance development if the cost magnitude is sufficiently large [38,39]. Previously, resistance to single-gene Bt crops carried minor or no detectable fitness costs [15,17,40–42]. In contrast, resistance to pyramided crops is more likely to be associated with fitness costs as it might evolve from mutations in more than one gene. The strength of the fitness costs may be magnified by ecological factors, such as host-plant quality to the insect herbivore [43,44]. Thus, information on the interaction of fitness costs of Bt resistance and the insect performance when feeding on different host plant species and varieties are informative for devising better resistance management [4,37,45].

Here we used a fall armyworm population resistant to Cry1A.105 + Cry2Ab2 Bt maize [9] to test the hypothesis of a positive association between two-toxin resistance and fitness costs and their increased magnitude by host crops less suitable for the herbivore. We compared life-history traits and population growth rates between Bt-resistant and -susceptible populations maintained on Bt and non-Bt cultivars of cotton, maize, and soybean to determine the potential pleiotropic effects of the resistance alleles. We found adverse effects on the fitness of the resistant insects for seven of the eight plant genotypes assessed, and the magnitude of the fitness costs was higher on host crops less suitable for the herbivore. These findings provide novel and exciting information helpful to refine current insect resistance management strategies for pyramided Bt crops.

## 2. Materials and Methods

### 2.1. Insect Populations

The susceptible fall armyworm population (Bahia-Cv) was collected in 2013 from Cry1F Bt maize fields in Luís Eduardo Magalhães, in Bahia, Brazil, and maintained in the laboratory without exposure to pesticides since its establishment [9]. No individual of the fall armyworm rice strain was found using molecular markers [46]. The resistant population (Bahia-Bt) was derived from the Bahia-Cv population by selecting for resistance to Cry1A.105+Cry2Ab2 Bt maize [9]; the resistance trait is recessive and confers cross-resistance to Cry1F but not to Vip3Aa Bt toxins [9,47]. The insects were reared using standard rearing conditions at $27 \pm 2\,°C$, $70 \pm 15\%$ r.h., and photoperiod of 14:10 h light: dark [48].

### 2.2. Plants, Growth Conditions, and Foliage for Assays

We used eight cultivars of the main Bt crops grown worldwide, including cotton, maize, and soybean [2]. Some cultivars express cry Bt transgenes, but not the selecting agents Cry1A.105 and Cry2Ab2 (Table 1). In the greenhouse, we planted five seeds of each cultivar in 14-L pots containing standard potting soil mixture. After two weeks, we thinned the seedlings to three plants per pot. We irrigated and fertilized the soil for optimal plant growth and used no pesticide for plant protection. We tested the expression of *cry1A* and *cry1F* transgenes in the plants using kits STX 10301/0050 or STX 06200/0050 according to the manufacturer's instructions (Agdia Inc., Elkhart, IN, USA). The plants used as the source of foliage for the experiment were in the mid-vegetative V4–V8 growth stages of maize and soybean and the early reproductive stages of cotton plants, between flowering and boll formation. We selected only healthy plants and excised the leaves from the maize whorl or the upper canopy of cotton and soybean. To help maintain turgidity, we immediately placed the leaf bases or petioles in a bucket with water to be taken to the laboratory until being placed in the experimental units.

**Table 1.** Non-Bt and Bt crop cultivars used in the study.

| Host Crop | Trait | Abbreviation | Cultivar, Transgenic Event, and Seed Company | Bt Toxin |
|---|---|---|---|---|
| Maize | Non-Bt | Non-Bt maize | 30F53, Isogenic hybrid, Corteva Agriscience | Non-Bt |
| Maize | Bt | Cry1F maize | 30F53H, TC1507, Corteva Agriscience | Cry1F |
| Maize | Bt | Cry1Ab maize | 30F53Y, MON810, Corteva Agriscience | Cry1Ab |
| Maize | Bt | Cry1F+Cry1Ab maize | 30F53HY, TC1507 × MON810, Corteva Agriscience | Cry1Ab, Cry1F |
| Soybean | Non-Bt | Non-Bt soybean | MSOY8866, Near-isogenic cultivar, Monsoy Bayer | Non-Bt |
| Soybean | Bt + Glyphosate resistance | Cry1Ac soybean | M8330IPRO, MON87701 × MON89788, Monsoy Bayer | Cry1Ac |
| Cotton | Non-Bt | Non-Bt cotton | Delta Opal, Isogenic cultivar), D&PL Brazil | Non-Bt |
| Cotton | Bt | Cry1Ac cotton | NuOpal MON531, D&PL Brazil | Cry1Ac |

The same abbreviations were used throughout the text, tables, and figures. Expression/non-expression of Bt proteins was confirmed using ELISA-based assays (EnviroLogix, Quantiplate kits, Portland, ME).

### 2.3. Life-History Costs in the Immature Stage

The eight plant genotypes and two insect populations were assayed in an entirely randomized longitudinal experiment, conducted in the same controlled conditions as previously described for rearing the insects. There were ten replications, each one consisting of a 16-well polyvinylchloride tray (Advento Pásticos, Diadema, São Paulo, Brazil). We assigned the foliage of each plant genotype to 20 trays, 10 for each insect population. The foliage placed in the tray well consisted of 3–4-cm sections of maize whorl leaves, half or two-thirds of a fully expanded cotton leaf, or leaflets of a trifoliate soybean. In total, there were 16 fall armyworms per replicate or tray, 160 per plant genotype, and 1280 of each population in the eight plant genotypes. We replaced the foliage every other day and recorded the survival rate until pupation. We also recorded the developmental time (i.e., duration of the larval stage). The pupae were weighed within 24 h after pupation, sexed, and transferred to plastic containers lined with paper-towel tissue misted with distilled water to avoid desiccation.

### 2.4. Fitness Costs to Population Growth

Using the pupae from the above cohorts according to plant genotype and insect population, we calculated the sex ratio and set twenty male-female pairs. Each pair was placed in a polyvinyl chloride cage (10 cm height × 10 cm Ø) and maintained at the previously described conditions. Moth food based on 10% honey solution in water was added to the cages. To estimate fertility, we recorded the number of egg masses laid by each female daily until the end of the egg-laying period (i.e., seven days). Egg masses were individually transferred to 200-mL plastic pots for hatching, and the number of neonates produced in each egg mass was recorded. The population growth potential was estimated using the life-table format [49,50]. Population growth parameters calculated were the net

reproductive rate (i.e., the number of times a population will multiply per generation, $R_0$), the generation time (the mean period between the birth of individuals of a generation and that of the next generation, $T$), and the intrinsic rate of population increase (daily female offspring production per parental female, $r_m$). More details on algorithms used for the calculations are in the data analysis section below.

### 2.5. Dominance of Fitness Costs

Reciprocal crosses between the resistant and susceptible populations (i.e., s♀ × r♂ and s♂ × r♀) were conducted to test whether the fitness costs were apparent on the heterozygotes when reared on non-Bt maize. The procedures for the crosses were the same as described elsewhere [9,48], and the population growth potential was determined as described above.

### 2.6. Ranking Host-Crop Cultivars That Magnified the Fitness Costs

To compare larval performance and host plant suitability, we calculated a 'fitness index' (*fi*) using the formula [51] $fi = lx \times mx \div ti$, where $lx$ is the percentage of pupation, $mx$ is mean pupal mass, and $ti$ is the duration of the immature stage. Using the index according to host-crop cultivar, we estimated the relative fitness of the Bt-resistant insects to the susceptible ones for each experimental replicate. To obtain insight into the strength of fitness costs to constrain the numbers of resistant insects in relation to susceptible ones, we simulated scenarios of fall armyworm population sizes using the intrinsic rate of population increase as estimated for the resistant and susceptible insects. We assumed that the population has exponential growth described in the model $Nt = N_0 \times e^{rm} \times t$, where $N_t$ is the size of the population at time t (e.g., generation 5, 10, 15, and 20), $N_0$ is the initial size of the population (initial number of individuals), and $r_m$ is the estimate of the intrinsic rate of increase [49] for each combination of population and cultivar. These procedures allowed us to estimate the percentage of population growth suppression of Bt-resistant individuals at generation t = 5, 10, 15, and 20, using the formula $\%PGsup = 100 \times (1 - (NtRES/NtSUS))$.

### 2.7. Statistical Analysis

Linear statistical modeling was used to compare life-history traits (i.e., survival, development time, and body size in the pupa stage) between resistant and susceptible armyworms for each host-crop cultivar. The data set for each response variable met the assumptions of normality and homoscedasticity, as indicated by residual analyses (*proc mixed* followed by *proc univariate* and *proc gplot*) [52]. Likewise, the relative fitness indexes of resistant insects in each host-crop cultivar were subjected to analysis of variance, and similar means were grouped using Scott-Knott post-hoc procedure ($p < 0.05$).

Population growth parameters and their associated variances were estimated [53,54] with suitable algorithms in SAS [55]. This procedure produced *p*-values for a one-tailed *t*-test on the hypothesized reduced fitness of resistant insects, which were adjusted using Bonferonni correction. In addition, path analysis using *proc calis* [52] to was used to test the hypothesized relationships between the effects of the environment without the selection agents (i.e., food plants lacking Cry1A.105+Cry2Ab2) on crucial life-history traits and life-table parameters of the fall armyworm populations.

## 3. Results

Fitness components of fall armyworms showing differential resistance to dual-gene Bt maize were measured in eight host-crop cultivars, and in seven of them, we observed substantial life-history costs to larval survival, development time, and body size (i.e., pupal weight) (Figures 1 and 2). No delay in development time or reduced survival of the resistant larvae was found when they were reared on Cry1Ab maize foliage (Figure 1C); on Cry1Ab+Cry1F maize, only development time was reduced (Figure 1D). Strikingly, only one resistant larva completed development on non-Bt cotton foliage (Figure 2A), showing its strong effect on the fitness costs of the resistance.

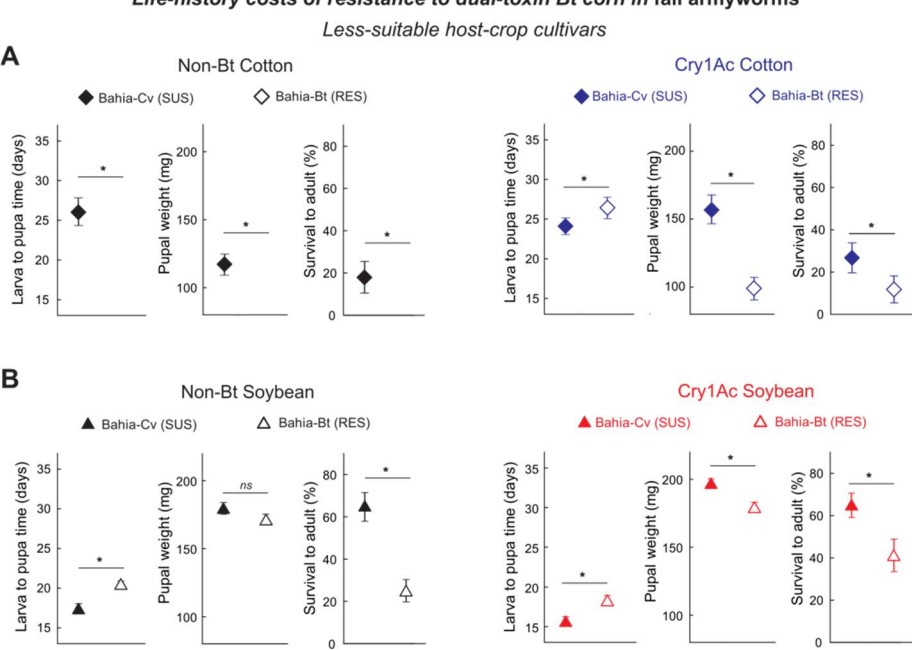

**Figure 1.** Life-history costs of resistance to dual-gene Bt maize (Cry1A.105+Cry2Ab2) in the invasive fall armyworm (*Spodoptera frugiperda*). Larvae (*n* = 160) of a Bt-resistant population (Bahia-Bt, black symbols) and susceptible population (Bahia-Cv, white symbols) were reared on the foliage of four maize cultivars (panels **A**–**D**), and their life-history traits were recorded. Data are means ± standard errors. Asterisks indicate significant differences (ANOVA, *p* < 0.05) between the insect populations. ns: Not Significant.

**Figure 2.** Influence of less-suitable host plants on the life-history costs of resistance to dual-gene Bt maize (Cry1A.105+Cry2Ab2) in the invasive fall armyworm (*Spodoptera frugiperda*). Larvae (*n* = 160) of a Bt-resistant population (Bahia-Bt, black symbols) and susceptible population (Bahia-Cv, white symbols) were reared on the foliage of cultivars of (**A**) cotton and (**B**) soybean, and their life-history traits were recorded. Data are means ± standard errors. Asterisks indicate significant differences (ANOVA, *p* < 0.05) between the insect populations. ns: Not Significant.

When comparing the population growth potential (i.e., population fitness), we found reduced net reproductive rate ($R_0$) and intrinsic rate of population increase ($r_m$), as well as longer generation time, for the resistant population on all host-crop cultivars, except Cry1Ab maize (Figure 3, Table 2). In contrast, the intrinsic rate of population increases of insects heterozygous for the resistance was not significantly different of the susceptible population, indicating that the fitness cost of the resistance was inherited as a recessive trait (Figure 4).

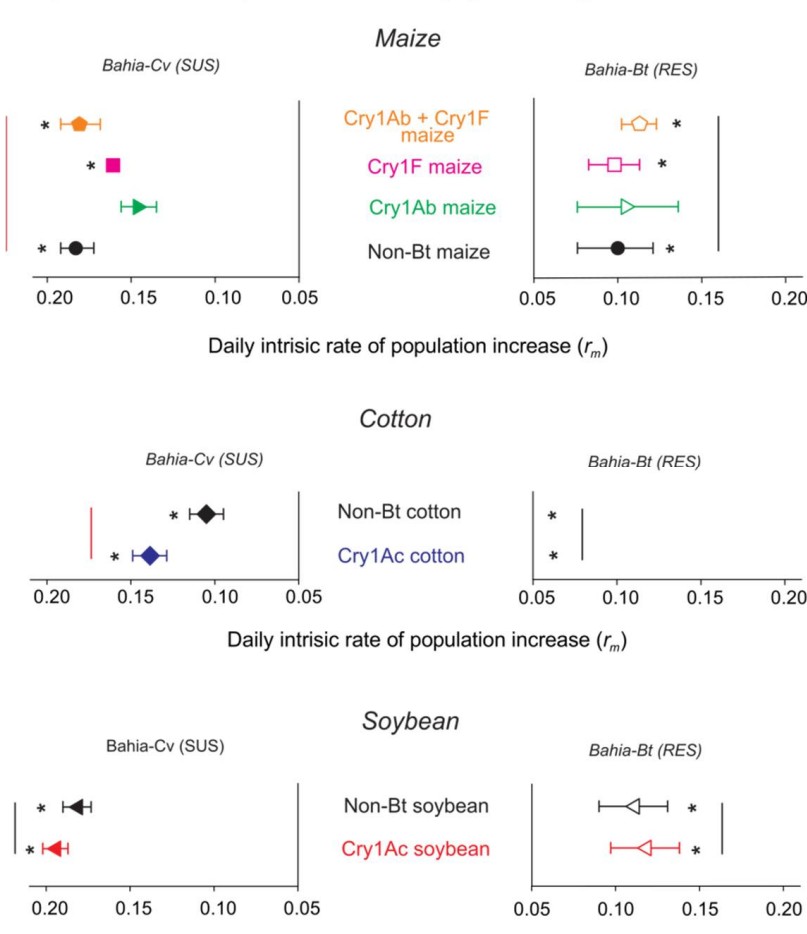

**Figure 3.** Population fitness costs of fall armyworm (*Spodoptera frugiperda*) resistance to dual-gene Bt maize (Cry1A.105+Cry2Ab2). Shown are intrinsic rates of population increase for Bt-resistant and -susceptible fall armyworms reared on the foliage of eight host-plant cultivars (n = 12–17 families per insect population per host-plant cultivar). Data are means and standard errors. * Reduced population fitness ($p < 0.05$), one-tailed *t*-test using jackknife-estimated variances. Values under the same vertical line are not significantly different ($p < 0.05$) using Bonferonni correction.

We hypothesized that low host-plant quality for fall armyworm larvae would magnify fitness costs of resistance. On non-Bt cotton and soybean as well as on Cry1Ac cotton, the resistant insects had relative fitness values lower than on Cry1Ab maize ($p < 0.05$), host-crop cultivar on which the costs were least apparent (Figure 5). In addition, both non-Bt and Bt cotton suppressed the population growth of the resistant individuals, whereas the other host-crop cultivars caused a lower suppressive effect, particularly Cry1Ab maize (Figure 6). Hence, the data supported our hypothesis and cultivars of cotton and soybean most magnified the fitness costs of resistance (Figures 5 and 6).

**Table 2.** Fitness costs of fall armyworm resistance to dual-gene Bt maize (Cry1A.105+Cry2Ab2). Shown are population growth statistics for Bt-resistant and -susceptible fall armyworms reared on the foliage of eight host-plant cultivars (*n* = 12–17 families per insect population per host-plant cultivar).

| Host-Plant Cultivar | Population Growth Parameter [1] | Fall Armyworm Population | | *p* |
| --- | --- | --- | --- | --- |
| | | Susceptible | Resistant | |
| Cry1Ac cotton | $r_m$ | 0.139 ± 0.010 | nd | nd |
| | $R_0$ | 72.1 ± 21.8 | nd | nd |
| | $T$ | 31.0 ± 1.2 | nd | nd |
| Non-Bt cotton | $r_m$ | 0.105 ± 0.010 | nd | nd |
| | $R_0$ | 25.1 ± 9.5 | nd | nd |
| | $T$ | 31.3 ± 0.8 | nd | nd |
| Non-Bt soybean | $r_m$ | 0.195 ± 0.007 | 0.118 ± 0.020 | <0.01 |
| | $R_0$ | 189.4 ± 35.6 | 22.2 ± 10.6 | <0.01 |
| | $T$ | 27.0 ± 0.3 | 27.3 ± 0.5 | 0.28 |
| Cry1Ac soybean | $r_m$ | 0.182 ± 0.008 | 0.111 ± 0.020 | <0.01 |
| | $R_0$ | 106.4 ± 22.3 | 21.4 ± 11.3 | <0.01 |
| | $T$ | 25.9 ± 0.2 | 29.1 ± 0.1 | <0.01 |
| Non-Bt maize | $r_m$ | 0.183 ± 0.009 | 0.100 ± 0.021 | <0.01 |
| | $R_0$ | 130.6 ± 34.1 | 15.5 ± 7.9 | <0.01 |
| | $T$ | 26.8 ± 0.5 | 28.1 ± 1.2 | 0.08 |
| Cry1Ab maize | $r_m$ | 0.146 ± 0.010 | 0.106 ± 0.030 | 0.08 |
| | $R_0$ | 48.8 ± 18.3 | 17.0 ± 11.7 | 0.07 |
| | $T$ | 27.0 ± 0.5 | 28.1 ± 0.9 | 0.12 |
| Cry1Ab+Cry1F maize | $r_m$ | 0.182 ± 0.010 | 0.113 ± 0.010 | <0.01 |
| | $R_0$ | 131.8 ± 32.5 | 24.0 ± 8.8 | <0.01 |
| | $T$ | 26.9 ± 0.4 | 28.6 ± 0.3 | <0.01 |
| Cry1F maize | $r_m$ | 0.161 ± 0.001 | 0.098 ± 0.015 | <0.01 |
| | $R_o$ | 79.2 ± 20.6 | 15.9 ± 6.2 | 0.01 |
| | $T$ | 27.7 ± 0.5 | 28.9 ± 1.2 | 0.13 |

[1] R0, net reproductive rate (the number of times the population multiplies per generation); rm, intrinsic rate of population increase (per day); T, mean generation time (days) (see Materials and Methods). nd, Not determined due to death or lack of reproduction of the resistant insects, indicating strong fitness costs of the resistance. Data are means ± SE and p values for one-tailed t-tests on the hypothesized reduced fitness (i.e., reduced rm and R0; increased T) of resistant insects.

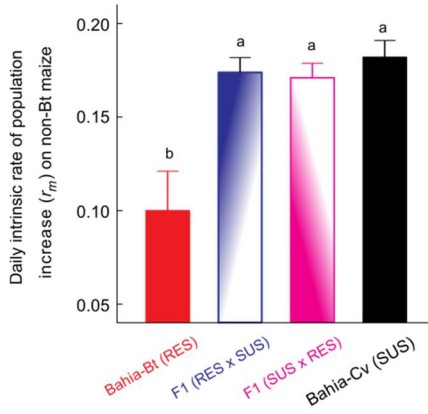

**Figure 4.** Dominance of the fitness costs associated with resistance to dual-gene Bt maize (Cry1A.105+Cry2Ab2) in the invasive fall armyworm (*Spodoptera frugiperda*). Shown are intrinsic rates of population increase ($r_m$) for the susceptible and resistant populations and for the offspring of their reciprocal crosses, all of them reared on leaves of non-Bt maize. Means ± standard errors followed by the same letter are not significantly different (*p* > 0.05, *t*-test using the jackknife estimate of variance with α correction using Bonferonni procedure).

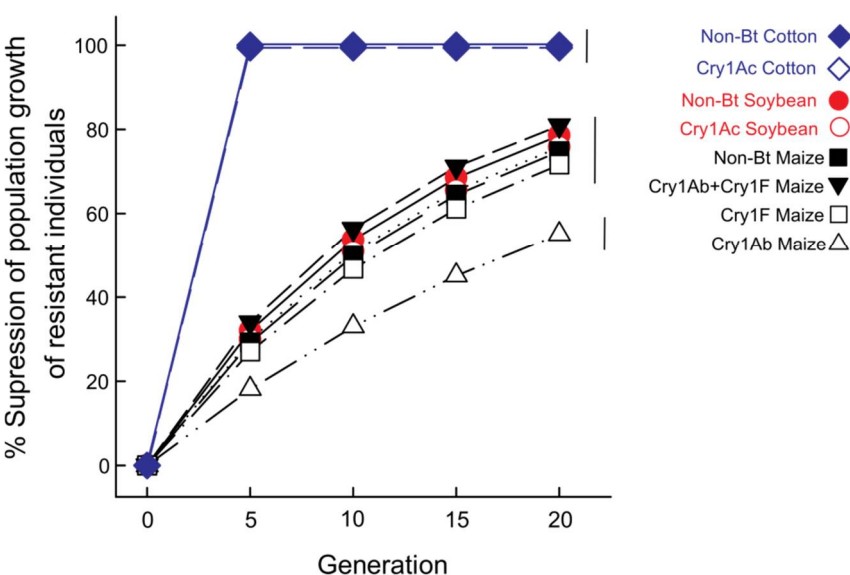

**Figure 5.** Effect of host-crop cultivars on the fitness costs of resistance to dual-gene Bt maize (Cry1A.105+Cry2Ab2) in the invasive fall armyworm (*Spodoptera frugiperda*). The relative fitness index was calculated by dividing the fitness index value of the Bt-resistant population by the fitness index value of the susceptible population. Means ± standard errors followed by the same letter are not significantly different (Scott-Knott test, $p > 0.05$).

**Figure 6.** Potential suppression of population growth of a Cry1A.105+Cry2Ab2-resistant population of the fall armyworm (*Spodoptera frugiperda*) in the foliage of eight host-plant cultivars due to fitness costs of the resistance. The population size ($N_t$) for the Bt-resistant and Bt-susceptible population was estimated using the exponential growth model for each cultivar; at generation *t*, the population size value of the Bt-resistant population was divided by the population size value of the Bt-susceptible population (see Section 2).

Larval fitness components that most influenced the population growth potential of fall armyworm were investigated using path analysis (Figure 7 and Table 3). We expected that environments without the selecting agents (i.e., host-plant foliage not producing Cry1A.105 and Cry2Ab2) directly influenced larval weight and developmental time determining pupal weight and survival, potentially leading to indirect effects on generation time and reproduction ($T$ and $R_0$). These life-table parameters, $T$ and $R_0$, would directly determine the population growth rate ($r_m$). The hypothesized path diagram was suitable, as indicated by the non-significant departures from expected covariance matrices ($\chi^2$ = 13.21, $df$ = 8, $p$ = 0.10). In the diagram, larval weight and developmental time correlated negatively. Larval weight positively affected pupal weight, while developmental time had the opposite effect. Developmental time also negatively affected survival and net reproductive rate ($R_0$) but positively affected generation time ($T$). Pupal weight had a direct, negative effect on generation time but no significant effect ($p$ > 0.05) on the net reproductive rate. The population growth potential (i.e., intrinsic rate of population increase, $r_m$) was affected mainly by the net reproductive rate, followed by the generation time with indirect contributions of survival, pupal (and larval) weight, and development time. This latter life-history trait is most affected by the other traits in the path diagram, as indicated by its higher total effects than the other variables (Figure 7, Table 3).

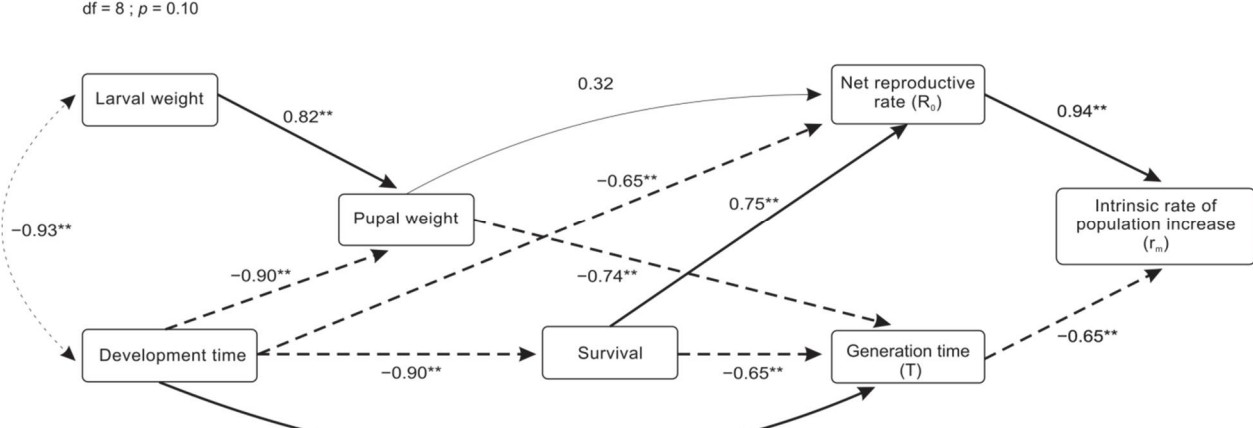

**Figure 7.** Path analysis diagram of the influence of Bt-toxin-free environment (i.e., maize, cotton, and soybean foliage) on life-history traits and population fitness of fall armyworm populations showing differential resistance to Cry1A.105+Cry2Ab2 Bt toxins. The result of $\chi^2$ goodness-of-fit for the path model is indicated. One-headed arrows indicate causal relationships (regression), while doubled-headed arrows indicate the correlation between the variables. Asterisks represent significance levels (** $p$ < 0.01), and the thickness of each line is proportional to the strength of the relationship. Solid arrows indicate positive relationships, while dashed arrows indicate negative relationships. Direct, indirect, and total values for path coefficients are shown in Table 3.

**Table 3.** Direct (D), indirect (I), and total (T) effects in the path diagram of Figure 7 for the model on the influence of resistance to Cry1A.105+Cry2Ab2 on life-history traits and their potential consequences for population growth of the fall armyworm.

| Variable | Pupal Weight | | | Survival | | | Net Reproductive Rate ($R_0$) | | | Generation Time ($T$) | | | Intrinsic Rate of Increase ($r_m$) | | |
|---|---|---|---|---|---|---|---|---|---|---|---|---|---|---|---|
| | D | I | T | D | I | T | D | I | T | D | I | T | D | I | T |
| Larval weight | −0.014 | - | −0.014 | - | 0.003 | 0.003 | - | 0.040 | 0.040 | - | $2 \times 10^{-4}$ | $2 \times 10^{-4}$ | - | $1.7 \times 10^{-5}$ | $1.7 \times 10^{-5}$ |
| Development time | −12.64 | - | −12.64 | −7.76 | 2.67 | −5.09 | −39.16 | 22.39 | −11.77 | 0.4 | 0.140 | 0.540 | - | $-9 \times 10^{-3}$ | $-9 \times 10^{-3}$ |
| Pupal weight | - | - | - | −0.21 | - | −0.21 | −2.26 | −0.25 | −2.51 | −0.1201 | −0.001 | −0.013 | - | $-1 \times 10^{-3}$ | $-1 \times 10^{-3}$ |
| Survival | - | - | - | - | - | - | 1.2 | - | 1.20 | 0.004 | - | $4 \times 10^{-3}$ | - | $6 \times 10^{-4}$ | $6 \times 10^{-4}$ |
| $R_0$ | - | - | - | - | - | - | - | - | - | - | - | - | $-5 \times 10^{-3}$ | - | $-5 \times 10^{-3}$ |
| $T$ | - | - | - | - | - | - | - | - | - | - | - | - | $5 \times 10^{-3}$ | - | $5 \times 10^{-3}$ |
| $R^2$ | 0.66 | | | 0.83 | | | 0.61 | | | 0.63 | | | 0.93 | | |
| $p$ | <0.01 | | | <0.01 | | | 0.02 | | | 0.01 | | | <0.01 | | |

## 4. Discussion

Using the fall armyworm as a model system to study insect resistance to a pyramided Bt crop, we have shown evidence of substantial fitness costs of resistance to Cry1A.105+Cry2Ab2-expressing Bt maize in seven host-plant cultivars of cotton, soybean, and maize. The life-history costs manifested as delayed developmental time and reduced survival rate and pupal weight. At the population level, the individual disadvantages translated into substantial costs to the net reproductive rate, generation time, and intrinsic rate of population increase. Strong fitness costs and incomplete resistance to dual-gene Bt maize [9] favor the durability of pyramided Bt crops [35,36,56,57]. In terms of evolutionary ecology applied to Bt crops, fitness costs increase the selection coefficient for susceptible individuals in refuges, and incomplete resistance reduces the fitness of resistant individuals on Bt crops [4,37,38]. Our findings help explain the relatively sustained efficacy of Cry1A.105+Cry2Ab2 maize against fall armyworms in Brazil. After more than a decade of the commercial release of the MON89034 transgenic maize event, which produces both Cry1A.105 and Cry2Ab2 toxins, no reduced efficacy or evolution of field resistance to this dual-gene Bt maize have been documented so far [34,58,59], despite the high risk of cross-resistance between Cry1 toxins concurrently used [8,9] and variable control efficacy depending on the situation [34,58–60]. Pyramided Bt cultivars provide crop protection against the pest complex and may help maintain their susceptibility to the Bt pesticidal proteins aided by the individual fitness penalties associated with the incomplete resistance that may be selected [4,6,9]. Pleiotropic effects of resistance alleles, as shown here, can impair the fitness of resistant insects, delaying the development of resistance or even restoring the susceptibility in the absence of selection pressure for resistance.

Because the fitness costs of multi-toxin resistance were enhanced on less-suitable host plants, refuges with specific host plants or cultivars may maximize fitness costs of resistance, which would increase the coefficient of selection for susceptible individuals in refuges [61], therefore, reducing the heritability of resistance [4,6,62]. This practice could work even better if the fitness costs are dominant [4,37] and merit further research to exploit the refuge as a reservoir of beneficial genes to slow the selection of insect resistance to Bt crops [43,61,63–65]. For example, when the resistant moths lay eggs on cotton plants, their offspring may not pass the resistance alleles to subsequent generations. Hence, cotton fields could help delay or prevent the spreading of the resistance alleles more efficiently than soybean and maize fields, deserving attention for its feasibility for resistance management of fall armyworm to Bt maize. In complex agricultural landscapes, such as in Brazil and the United States, soybeans could also help attract fall armyworm females to lay eggs in crops that offer additional challenges for pest development. Behavioral manipulation and environmental stressors, such as temperature and biotic interactions [66,67] should also be investigated to identify and quantify factors that may exaggerate fitness costs of fall armyworm resistance to Bt pesticidal proteins used in pest management.

Developmental time played a significant role in the fitness profile of the armyworm, negatively affecting pupal weight, net reproductive rate, survival, generation time, and the intrinsic rate of population increase of the fall armyworm. Mechanistically, delayed development in Bt-resistant insects may be caused by mutations in receptor proteins affecting nutrient uptake [68]. In the field settings, more extended development affects the rate of resistance evolution depending on complex interactions [69], including increased risk of natural mortality, decreased population growth rates, and increased likelihood of assortative mating. Most resistance management strategies assume random mating between resistant and susceptible moths [6]. We found that susceptible adults emerged 3–4 days before resistant ones, and developmental time was the life-history trait that accounted for most of the fitness variation of the armyworm. More research is needed to determine whether developmental asynchrony does not compromise the random mating assumption [6] and, consequently, the efficacy of refuge for resistance management [40,70].

The variation in the expression of the fitness costs of dual-toxin Bt resistance across host plants seems minor than that observed for single-toxin resistance [37,38,63]. This smaller

phenotypic plasticity (apparent independently of other variables) may be related to the high strength of the costs, which were consistently observed in various rearing environments (i.e., host-plant cultivars), contrasting with the more variable costs of insect resistance to single Bt toxin [37,43,63,64,71,72]. Notably, fitness costs more visible to selection independently of the ecological conditions are more valuable for resistance management [73].

The host-crop cultivar influenced the magnitude of the fitness costs; it was lower on Cry1Ac cotton than on the non-Bt cultivar, a pattern observed for soybean but not for Cry1Ab maize. These results are likely associated with low inherent susceptibility of fall armyworm to Cry1A Bt toxins [30,31,69,74,75], although stimulatory effects of sublethal-toxin exposure and differences in plant quality for the herbivore cannot be ruled out [76,77]. The resistance costs were lower for the fall armyworms on the foliage of maize and soybean than cotton. Maize seems to be the optimal (i.e., preferred) host plant of *S. frugiperda* in relation to soybean and cotton, on which the resistant insects had the lowest fitness indices and null life-table statistics (i.e., cohort extinction). Low quality of cotton foliage for fall armyworm larvae were previously reported [40] and may be due to gossypol, a phenolic compound toxic to many insect herbivores [78–80] that can affect many life-history traits [75] and amplify the fitness costs of Bt resistance [43,64,71,81]. Mutations associated with Bt resistance may interfere with the functional or structural integrity of the insect gut [37,40,43,68,82], favoring gossypol penetration and its adverse effects [43].

Our findings contrast with the lack of fitness costs of fall armyworm resistance to single-gene Bt maize or the singular pesticidal proteins Cry1A.105 and Cry2Ab2 [15,40–42,83,84] but are consistent with the prediction of greater costs of multiple-toxin resistance [4,82]. Our findings partially agree with those reported for fall armyworms selected for resistance to Bt maize producing Cry1A.105, Cry1F, and Cry2Ab2 after $F_2$-screen procedures [11,85]. In such studies, however, the resistance trait carried fitness costs of magnitude lower than the substantial fitness costs we observed here. In this study, the resistant population was field derived from collections in 2013 in a region of intensive agriculture, with complaints of control failure of fall armyworm by Cry1F maize and concurrent Cry1A.105+Cry2Ab2 maize adoption. When we established the base population in the laboratory, the larval survival rate on Cry1A.105+Cry2Ab2 maize was above 20% [9], indicating its past field-selection, which must have been aggravated by the positive cross-resistance between Cry1 Bt toxins in fall armyworm [8,9,47,57].

Two distinct mutant alleles likely confer the resistance to Cry1A.105 and Cry2Ab2 in the fall armyworm. Cry1 and Cry2 pesticidal proteins seem to bind to different sites and have distinct receptor-mediated resistance mechanisms, with no cross-resistance documented so far [8,9,86–88]. Lack of fitness costs of single-gene resistance to Cry1A.105 and Cry2Ab2 have been described [83,84]. In contrast, there were measurable fitness costs when the fall armyworms carried resistance alleles to both Cry1A.105 (or Cry1F) and Cry2Ab2 [11,85]. This observation is in line with the hypothesis that the higher the number of mutated resistance genes, the higher the expected fitness costs, owing to pleiotropic effects [4,82], although there could be the selection of less costly alleles or fitness modifying genes that may mitigate the fitness costs [89]. We identified substantial fitness costs when the fall armyworms were reared in different Bt cultivars, and in some of them, the strong fitness costs were magnified even more, suggesting favorable conditions for managing resistance of fall armyworm to Bt crops.

In summary, this research shows that the Cry1A.105+Cry2Ab2 resistance allele(s) have strong pleiotropic effects, reducing the fitness in the resistant fall armyworms in the absence of the pesticide proteins. Developmental time was the life-history trait accounting for most of the insect fitness profile. The lower-quality host-crop cultivars of cotton and soybean enhanced the fitness differential between the resistant and susceptible insects, highlighting host-plant quality as a significant modulating stressor affecting insect resistance management in the agroecosystem. If the substantial fitness costs of resistance to multi-toxin Bt maize occur in the field populations of fall armyworm, the costs may enhance the effect of

the refuge and gene pyramiding strategies to delay genetic adaptation to Bt crops by this notorious pest of worldwide importance in agriculture.

**Author Contributions:** Conceptualization, O.F.S.-A. and E.J.G.P.; Methodology, O.F.S.-A., C.S.T., J.V.C.R. and E.J.G.P.; Software, E.E.O. and R.N.C.G.; Validation, O.F.S.-A. and C.S.T. Formal Analysis, O.F.S.-A. Investigation, O.F.S.-A. Resources, E.E.O., E.J.G.P. and R.N.C.G.; Data Curation, O.F.S.-A. and E.J.G.P.; Writing—Original Draft Preparation, O.F.S.-A.; Writing—Review & Editing, E.J.G.P., E.E.O. and R.N.C.G.; Visualization, E.E.O.; Supervision, E.J.G.P.; Project Administration, E.J.G.P.; Funding Acquisition, E.J.G.P. and E.E.O. All authors have read and agreed to the published version of the manuscript.

**Funding:** Financial support was provided by the National Council of Scientific and Technological Development (CNPq; ID#: 430701/2016-0; 150434/2017-0) from the Brazilian Ministry of Science and Technology, the CAPES Foundation from the Brazilian Ministry of Education (Finance code 001), and the Minas Gerais State Foundation for Research Aid (FAPEMIG).

**Institutional Review Board Statement:** Not applicable.

**Informed Consent Statement:** Not applicable.

**Data Availability Statement:** The data that support the findings of this study are available on request from the corresponding author. The data are not publicly available due to privacy or ethical restrictions.

**Acknowledgments:** We thank the research assistants in the Laboratory of Insect-Plant Interactions for assisting with plant cultivation, insect rearing, and lab maintenance. The thoughtful comments of two anonymous reviewers helped improve the manuscript.

**Conflicts of Interest:** The authors declare that they have no conflict of interest.

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
