# Peer review of "Strong Fitness Costs of Fall Armyworm Resistance to Dual-Gene Bt Maize Are Magnified on Less-Suitable Host-Crop Cultivars"

_agronomy, doi:10.3390/agronomy12030682_

Round 1

Reviewer 1 Report

I read the study by Santos-Amaya et al. with a great interest. I believe that the results worth publishing. Without exaggeration, I can say that this is the best manuscript I have ever reviewed. However, I have several minor comments listed below that I would like to see addressed by the authors.

Firstly, I miss numbering of lines. Luckily, I do not need much because your manuscript is very well written.

I think this should be not in Introduction: We found adverse effects on the fitness of the resistant insects for seven of the eight plant genotypes assessed, and the magnitude of the fitness costs was higher on host crops less suitable for the herbivore. These findings provide novel and exciting information helpful to refine current insect resistance management strategies for pyramided Bt crops.

2.2. Plants, growth conditions, and foliage for assays: At the end of paragraph: To help maintain turgidity

Table 1 and Table 2: second line should be not in bold

Table 1: fifth line: in column Abbrevation you have „Cry1F/Cry1Ab maize“ but in column Bt toxin you do have only Cry1F. I think there should be also Cry1Ab.

Figure 3: Capital in headings: Impact of host-crop… BTW, I think these headings of figures are not suitable in scientific papers (all figures).

Figure 3: capital letter „Cry1Ab“ in part with corn graphs

Figure 3: differencas (note in Spanish)

Figure 3: *Reduced population fitness: Do you compare just non-Bt with other Bt host plants? I think you should specify it in figure legend. Moreover I think you shloud not use t-test but ANOVA for corn of t-test with Bonferroni correction of p-level. Did you do several t-tests (non-Bt x Cry1F, non-Bt x Cry1Ab, and so on?). You should use Bonferonni correction for p level. It is similar as on Figure 4.

Table 2: How you can statisticaly compare suspectible vs. resistant when population grow parameters are not determinded? How you estimated „hypothesized reduced fitness“.

Figure 6: Symbols for non-Bt cotton and Cry1Ac cotton are overlapping. I understand your intention but you should at least do bigger symbols fro Cry1Ac or add an explanation of the connecting lines of individual treatments.

Table 3: End of the heading is under the table.

Table 3: Formating: IE in column for pupal weight; DE and TE for survival.

End of first paragraph in Discussion: When we established the base population in the laboratory, the larval survival rate on Cry1A.105+Cry2Ab2 maize was above 20% [9], indicating its past fieldselection, which must have been aggravated by the positive cross-resistance between Cry1 Bt toxins in fall armyworm [9,47,65].        These larvae were considered as suspectible? You use this culture as control, are you? It means that fitness of control larvae could be influenced by this resistance? Can it be possible by their natural resistance to Cry proteins? I think genus Spodoptera is generally quite resistant against Cry proteins.

These two statements in Bullet list of main findings

  • We examined fitness costs of resistance to Cry1A.105+Cry2Ab Bt maize in fall armyworms of similar genetic background;
  • We compared insect life-history traits and population growth rates on foliage of Bt or non-Bt cultivars of maize, soybean, and cotton;

are not findings but description of your methods.

Author Response

Point 1: I read the study by Santos-Amaya et al. with a great interest. I believe that the results worth publishing. Without exaggeration, I can say that this is the best manuscript I have ever reviewed. However, I have several minor comments listed below that I would like to see addressed by the authors.

Reply: Thank you for the positive feedback. We appreciated the time and effort the referee took to review our article.

Point 2: Firstly, I miss numbering of lines. Luckily, I do not need much because your manuscript is very well written.

REPLY: The numbering of lines was provided in the revised version.

Point 3: I think this should be not in Introduction: We found adverse effects on the fitness of the resistant insects for seven of the eight plant genotypes assessed, and the magnitude of the fitness costs was higher on host crops less suitable for the herbivore. These findings provide novel and exciting information helpful to refine current insect resistance management strategies for pyramided Bt crops.

REPLY: We respectfully disagree with referee's opinion. I believe these sentences maintain the reader focused on the main findings of manuscript. I have decided to maintain this text segment.

Point 4: 2.2. Plants, growth conditions, and foliage for assays: At the end of paragraph: To help maintain turgidity

Table 1 and Table 2: second line should be not in bold

REPLY: Fixed.

Point 5: Table 1: fifth line: in column Abbrevation you have "Cry1F/Cry1Ab maize "but in column Bt toxin you do have only Cry1F. I think there should be also Cry1Ab.

REPLY: That is right. It was fixed as suggested and updated accordingly.

Point 6: Figure 3: Capital in headings: Impact of host-crop… BTW, I think these headings of figures are not suitable in scientific papers (all figures).

REPLY: That is right. Fixed as suggested.

Point 7: Figure 3: capital letter "Cry1Ab "in part with corn graphs

REPLY: That is I right. Fixed as suggested.

Point 9: Figure 3: differencas (note in Spanish)

REPLY: That is right. Fixed as suggested.

Point 10: Figure 3: *Reduced population fitness: Do you compare just non-Bt with other Bt host plants? I think you should specify it in figure legend. Moreover I think you shloud not use t-test but ANOVA for corn of t-test with Bonferroni correction of p-level. Did you do several t-tests (non-Bt x Cry1F, non-Bt x Cry1Ab, and so on?). You should use Bonferonni correction for p level. It is similar as on Figure 4.

REPLY: We have used the correction as suggested. For Fig. 3 we have inserted: "Values under the same vertical line are not significantly different (P < 0.05) using Bonferonni correction." Likewise for Fig. 4: "with a correction using Bonferonni procedure)."

Point 11: Table 2: How you can statisticaly compare suspectible vs. resistant when population grow parameters are not determinded? How you estimated "hypothesized reduced fitness ".

REPLY: That's right. We have replaced the information with "nd, not determined".

Point 12: Figure 6: Symbols for non-Bt cotton and Cry1Ac cotton are overlapping. I understand your intention but you should at least do bigger symbols fro Cry1Ac or add an explanation of the connecting lines of individual treatments.

REPLY: That's right. We have inserted a new legend and clarified the figure caption.

Point 13: Table 3: End of the heading is under the table

REPLY: Fixed as suggested.

Point 14: Table 3: Formating: IE in column for pupal weight; DE and TE for survival.

REPLY: Fixed as suggested.

Point 15: End of first paragraph in Discussion: When we established the base population in the laboratory, the larval survival rate on Cry1A.105+Cry2Ab2 maize was above 20% [9], indicating its past field selection, which must have been aggravated by the positive cross-resistance between Cry1 Bt toxins in fall armyworm [9,47,65].        These larvae were considered as susceptible? You use this culture as control, are you? It means that fitness of control larvae could be influenced by this resistance? Can it be possible by their natural resistance to Cry proteins? I think genus Spodoptera is generally quite resistant against Cry proteins.

REPLY: We understand the reviewer's concern. Spodoptera is generally quite tolerant to Cry proteins. Nevertheless, the combined effects of Cry1A + Cry2A Bt maize were highly efficacious against the fall armyworm larvae when they were commercially released (2010). The frequency of less-susceptible individuals in the Brazilian fall armyworm populations of 2011 and 2012 was relatively low, with Cry1A.105+Cry2Ab2 maize killing more than 95% of the larvae [e.g., see Leite, (2012), p. 16)]. N. A. Leite, "Selection and characterization of resistance in a strain of Spodoptera frugiperda to transgenic corn expressing Cry1F (in Portuguese)", Federal University of Viçosa. (2012). Available in: https://www.locus.ufv.br/bitstream/123456789/26641/1/texto%20completo.pdf.

We used the field-collected larvae as a comparator. That was to control for genetic background by using a population (Bahia-CV) of similar origin as the resistant population (Bahia-Bt). We considered Bahia-CV as "susceptible" in relation to the resistant population. This was further selected in the laboratory to amplify and purify the resistance. This trait already existed in a few individuals in the base population. Given Bt proteins' complex mechanism of action, it is difficult to know whether the background tolerance to Cry1 and Cry2 Bt proteins in fall armyworm populations is associated with fitness costs, particularly in the more-tolerant individuals likely carrying resistance alleles. Unfortunately, studies on the baseline susceptibility or estimates of initial Cry1 and Cry2 resistance allele frequency were not conducted. Had we had the results of such studies, we would be in better position to guess whether or not fitness costs were present and helping selecting against resistance (tolerance).

Point 16: These two statements in Bullet list of main findings

We examined fitness costs of resistance to Cry1A.105+Cry2Ab Bt maize in fall armyworms of similar genetic background;

We compared insect life-history traits and population growth rates on foliage of Bt or non-Bt cultivars of maize, soybean, and cotton;

are not findings but description of your methods.

REPLY: The reviewer is correct. The bullet list was not mandatory. We have revised it accordingly as a wrap-up abstract containing some information on the study background and results.

Reviewer 2 Report

I thoroughly enjoyed this manuscript.  It is a different and well thought out piece of research that -like the great majority- is not pointing to doomsday scenarios with resistance to GE crops.

Of great significance and novel approach in the manuscript is their proposal that “…when the resistant moths lay eggs on cotton plants, their offspring may not pass the resistance alleles to subsequent generations. Hence, cotton fields nearby or rotated -I add, not necessary- with Bt maize fields could help delay or prevent the spreading of the resistance alleles more efficiently than soybean and maize fields, deserving attention on its feasibility for resistance management of fall armyworm to Bt maize.”  I completely agree with this statement, adding that in complex agricultural landscapes such as in Brazil and the United States, also soybeans would help attracting fall armyworm females to lay eggs in crops that offer additional challenges for pest development.  I believe this should be part of the abstract.

I also liked their interpretation that “…susceptible adults emerged 3-4 days before resistant ones, and more research is needed to determine whether larval developmental delays do not compromise the random mating assumption, and consequently the efficacy of refuge for resistance management”.  Even though this has been mentioned before, this other valuable contribution appears way at the end of the manuscript.  No wonder most pest models do not consider this quite important biological fact.  I encourage the authors to build a stronger -and more ‘visible’- case for this important statement.

Other than those two major and positive comments, I find the manuscript well organized, superb illustrations, specially figures 1-3, but honestly I spent quite a bit of time in figure 7 without obtaining a good ‘overall picture. Figure 6 should be corrected in the part of ‘fallworm’ and perhaps table 3 is unnecessary because it repeats the values given elsewhere.

As mentioned in the discussion, one of the strains involved in this research is resistant to Cry1A.105+Cry2Ab2 expressed by corn, and individual response (presence of resistant alleles) to individual proteins was not included or mentioned in the manuscript.  I don’t see a problem here but, it should be emphasized in the abstract that the FAW used are resistant to Cry1A.105+Cry2Ab2 expressing corn.  Right now it gives the impression that indeed this FAW strain is resistant to both proteins, which we don’t know for sure.

In the abstract “seven of the eight host crops studied” should be corrected.  Eight cultivars of three host crops is what this manuscript covers.

In page 6 the following statement appears “the resistance was similar to that of the”.  I believe a scientifically valid description for this should be ‘…was not significantly different…’

In figure 3 I believe ‘diferenças ???’ should be removed.

Lastly, “After seven years of the commercial release of the MON89034 event, which produces both Cry1A.105 and Cry2Ab2 toxins, no reduced efficacy or evolution of field resistance to this dual-gene Bt maize have been documented so far, despite the high risk of cross-resistance between Cry1 toxins concurrently used and variable control efficacy depending on the situation” deserves a better place in the manuscript and more emphasis.  This really complements what this well-written manuscript is all about.

Well done job!

Author Response

Point 1: I thoroughly enjoyed this manuscript. It is a different and well thought out piece of research that -like the great majority- is not pointing to doomsday scenarios with resistance to GE crops.

REPLY: We are glad that the reviewer liked our manuscript. We appreciated the time and efforts taken to review our article.

Point 2: Of great significance and novel approach in the manuscript is their proposal that "…when the resistant moths lay eggs on cotton plants, their offspring may not pass the resistance alleles to subsequent generations. Hence, cotton fields nearby or rotated -I add, not necessary- with Bt maize fields could help delay or prevent the spreading of the resistance alleles more efficiently than soybean and maize fields, deserving attention on its feasibility for resistance management of fall armyworm to Bt maize." I completely agree with this statement, adding that "in complex agricultural landscapes such as in Brazil and the United States, also soybeans would help attracting fall armyworm females to lay eggs in crops that offer additional challenges for pest development". I believe this should be part of the abstract.

REPLY: Thanks for the positive feedback. We have revised the entire text of the Discussion, deleting 'nearby or rotated with Bt maize fields' [L363] and inserting the suggested point. To the Abstract, we have some insertions, hoping to convey the piece of information highlighted by the reviewer. The changes are highlighted in yellow.

Point 3: I also liked their interpretation that "…susceptible adults emerged 3-4 days before resistant ones, and more research is needed to determine whether larval developmental delays do not compromise the random mating assumption, and consequently the efficacy of refuge for resistance management". Even though this has been mentioned before, this other valuable contribution appears way at the end of the manuscript. No wonder most pest models do not consider this quite important biological fact. I encourage the authors to build a stronger -and more 'visible'- case for this important statement.

REPLY: Thanks for the suggestion. We have revised the text, trying to highlight the point. [L393, 395–6]

Point 4: Other than those two major and positive comments, I find the manuscript well organized, superb illustrations, specially figures 1-3, but honestly I spent quite a bit of time in figure 7 without obtaining a good 'overall picture. Figure 6 should be corrected in the part of 'fallworm' and perhaps table 3 is unnecessary because it repeats the values given elsewhere.

REPLY: Again thanks for the positive comments, suggestions, and the editorial details pointed out in the figures. We have revised them accordingly, fixing the typos. Despite the relatively minor contribution of Figure 7 and Table 3, we decided to keep them in the revised version. The latter was designed to be a supplementary material (if the reader wanted to check the numerical values of the life-table parameters). However, given the online format used by MDPI, it would be best as part of the electronic article.

Point 5: As mentioned in the discussion, one of the strains involved in this research is resistant to Cry1A.105+Cry2Ab2 expressed by corn, and individual response (presence of resistant alleles) to individual proteins was not included or mentioned in the manuscript. I don't see a problem here but, it should be emphasized in the abstract that the FAW used are resistant to Cry1A.105+Cry2Ab2 expressing corn. Right now it gives the impression that indeed this FAW strain is resistant to both proteins, which we don't know for sure.

REPLY: That is right. We have revised the Abstract, trying to highlight the point. [L22]

Point 6: In the abstract "seven of the eight host crops studied" should be corrected. Eight cultivars of three host crops is what this manuscript covers.

REPLY: We have revised the point in the Abstract  [L18]

Point 7: In page 6 the following statement appears "the resistance was similar to that of the". I believe a scientifically valid description for this should be '…was not significantly different…'

REPLY: That is right. We have revised the text accordingly [L230].

Point 8:  In figure 3 I believe 'diferenças ???' should be removed.

REPLY: We have revised the figure accordingly.

Point 9:  Lastly, "After seven years of the commercial release of the MON89034 event, which produces both Cry1A.105 and Cry2Ab2 toxins, no reduced efficacy or evolution of field resistance to this dual-gene Bt maize have been documented so far, despite the high risk of cross-resistance between Cry1 toxins concurrently used and variable control efficacy depending on the situation" deserves a better place in the manuscript and more emphasis. This really complements what this well-written manuscript is all about.

REPLY: That is right. We have made a few revisions in the text, reorganizing  the tesxt of the Discussion. We have modified the sequence of the paragraphs to hit in the main points first and then in the secondary ones,  endind up with a summary paragraph. We believe all important points were discussed in better locations that a diligent reader easily can assess.

Well done job!

REPLY: Thanks again! 
